# The Amazon's 2023 Drought: Sentinel-1 Reveals Extreme Rio Negro River Contraction

**Fabien H. Wagner** [1,2,3,*] , **Samuel Favrichon** [2,3], **Ricardo Dalagnol** [1,2,3], **Mayumi C. M. Hirye** [1,3,4], **Adugna Mullissa** [1,3] and **Sassan Saatchi** [1,2,3]

1    Institute of the Environment and Sustainability, University of California, Los Angeles, CA 90095, USA; rdalagnol@ctrees.org (R.D.); ma.hirye@alumni.usp.br (M.C.M.H.); amullissa@ucla.edu (A.M.); sasan.s.saatchi@jpl.nasa.gov (S.S.)
2    NASA-Jet Propulsion Laboratory, California Institute of Technology, Pasadena, CA 91105, USA; samuel.favrichon@jpl.nasa.gov
3    CTREES.org, Pasadena, CA 91105, USA
4    Quapá Lab, Faculty of Architecture and Urbanism, University of São Paulo—USP, São Paulo 05508-900, SP, Brazil
*    Correspondence: wagner.h.fabien@gmail.com

**Abstract:** The Amazon, the world's largest rainforest, faces a severe historic drought. The Rio Negro River, one of the major Amazon River tributaries, reached its lowest level in a century in October 2023. Here, we used a U-net deep learning model to map water surfaces in the Rio Negro River basin every 12 days in 2022 and 2023 using 10 m spatial resolution Sentinel-1 satellite radar images. The accuracy of the water surface model was high, with an F1-score of 0.93. A 12-day mosaic time series of the water surface was generated from the Sentinel-1 prediction. The water surface mask demonstrated relatively consistent agreement with the global surface water (GSW) product from the Joint Research Centre (F1-score: 0.708) and with the Brazilian MapBiomas Water initiative (F1-score: 0.686). The main errors of the map were omission errors in flooded woodland, in flooded shrub, and because of clouds. Rio Negro water surfaces reached their lowest level around the 25th of November 2023 and were reduced to 68.1% (9559.9 km$^2$) of the maximum water surfaces observed in the period 2022–2023 (14,036.3 km$^2$). Synthetic aperture radar (SAR) data, in conjunction with deep learning techniques, can significantly improve near-real-time mapping of water surfaces in tropical regions.

**Keywords:** surface water mapping; synthetic aperture radar; deep learning; U-net; image segmentation; near-real time





## 1. Introduction

The Rio Negro River is among the largest tributaries of the left bank of the Amazon River, extending from its sources in Colombia to Venezuela and Brazil, making the Amazon the largest river basin in the world. The Rio Negro River has a length of 2230 km, a catchment area of 696,000 km$^2$, and an average flow of 28,400 m$^3$ per second, representing 14% of the annual average flow of the Amazon basin [1,2]. It participates in the complex hydrological processes of the Amazon River that play a key role in the water, energy, and carbon cycles and interact with the global climate system [3]. On the 26th of October 2023, the Rio Negro River water level measured at the Port of Manaus (Brazil) reached its lowest point since 1903, 12.7 m. The water level measurement at the port is currently the most reliable data for the Rio Negro River, and to date, no near-real-time data of the river height or extent derived from medium spatial resolution satellite observation (5–30 m) are easily available over the Rio Negro basin.

Generating maps of near-real-time water surface locations and extents for the Amazon region remains challenging. The primary obstacle is the persistent cloud cover. Despite the availability of numerous medium-spatial-resolution multispectral satellites like Landsat,

CBERS, Sentinel-2, or PlanetScope, which acquire data at frequencies of ≤once per month, the extensive cloud cover impedes the creation of near-real-time observations of the state of the river basin. As a result, most water surface products are generated based on a yearly or multi-annual frequency [4–8].

Only synthetic aperture radar (SAR), due to its cloud-penetrating capability, appears to be the data collection method of choice for the near-real-time monitoring of surface water in tropical regions with persistent clouds. Moreover, water surfaces of the Amazon River in synthetic aperture radar (SAR) images are easily distinguishable from other land cover types due to their characteristic lower backscattering values compared to most terrestrial surfaces. These lower backscattering values are likely attributed to the specular reflection and smoothness of the water [9,10].

There is a significant body of literature on the study of water levels and extent and wetland vegetation in the Amazon region using radar, employing various instruments (SIR-C, RADARSAT-2, ALOS PALSAR, and ALOS2 ScanSAR, PolSAR, ENVISAT, JERS-1), diverse bands such as X-, C-, and L-bands, and different polarizations [5,9–17], as reviewed in [3,8]. However, none of these studies provided near-real-time and high-resolution water surfaces because the instruments or missions were not designed for this purpose. The Sentinel-1 C-band SAR satellite of the European Space Agency (ESA), with its instrument operating at a center frequency of 5.405 GHz, enables image acquisition even in cloudy conditions. Furthermore, with a spatial resolution of 10 m, a temporal resolution of 12 days, and an open data policy, it currently appears to be the optimal satellite program for measuring near-real-time variations in water surfaces at a fine scale and has not been fully exploited in the Amazon [3].

To map water surfaces with Sentinel-1, several traditional methods are available, such as spectral indices [18], machine learning [19–21], dynamic thresholding [22–26], and neural networks [27]. However, recently, the remote sensing scientific community has been adapting to modern deep learning approaches [28]. For instance, the U-Net convolutional neural network can accurately map surface water from Sentinel-1 in tropical regions, achieving high validation accuracy, with an F1-Score exceeding 0.92 [28,29]. Another study, which provides a substantial dataset of flooding samples for training models, demonstrates that a fully convolutional (FCNN) model outperforms thresholding algorithms in identifying flooded areas [30]. Further research indicates that the U-Net algorithm can learn features as meaningful as spectral indices for flood segmentation [31]. In an attempt to map floods with CNN-based methods and Sentinel-1 imagery with minimal pre-processing, it was shown that CNNs, including the U-Net algorithm, can reduce the time required to develop a flood map by 80%, while maintaining strong performance across various locations and environmental conditions [32]. Lastly, for near-real-time flood mapping using Sentinel-1, the U-Net model demonstrated a notable improvement over thresholding techniques and enabled fast processing (1 min per image) with a very low omission error [33]. Therefore, we used the U-Net convolutional neural network [29] to map water surfaces in Sentinel-1 images, as this algorithm is currently one of the most successful for mapping water in Sentinel-1 images and is an easy-to-use algorithm for binary segmentation tasks [31].

Several products covering the Rio Negro River have been developed in the last decade. One of the most renowned and accurate water surface products is global surface water (GSW) at a 30 m resolution from the Joint Research Centre (JRC) [4]. This dataset used the entire multi-temporal orthorectified Landsat 5, 7, and 8 archives spanning the past 32 years (1984–2016) and employed machine learning techniques to quantify global surface water and its changes at a 30 m spatial resolution [4]. Specifically, using Google Earth Engine [34], they mapped the water surface in three million Landsat satellite images, identifying when water was present, how its occurrence changed, and the nature of changes in terms of seasonality and persistence. The main limitation of this product is that it is static and not available for recent years, and thus, cannot be used to monitor drought episodes in near-real time. Another remarkable static dataset for the region is the LBA-ECO LC-07 dataset of wetland extent for the Lowland Amazon Basin [5,35]. This dataset was derived

from mosaics of Japanese Earth Resources Satellite (JERS-1) synthetic aperture radar (SAR) imagery for the period October–November 1995 and May–July 1996. It provides a map of the wetland extent, vegetation type, and dual-season flooding state of the entire lowland Amazon basin at 3 arc-seconds of spatial resolution (∼90 m). Finally, the most up-to-date and comparable-to-near-real-time dataset is that for annual water surface from the MapBiomas initiative [6,7,36], particularly from the year 2022. The water surface is mapped by the MapBiomas initiative using machine learning techniques and the Landsat Data Archive covering the period from 1985 to 2022 (scenes with cloud cover <70%), available in the Google Earth Engine (Landsat 5 TM, Landsat 7 ETM+, and Landsat 8 OLI). This dataset is produced with >84% average user accuracy [7,36]. A limitation of the product is that the map is produced yearly, so it cannot be used for near-real-time mapping, for example, for the October–November 2023 drought. While these three water surface datasets have limitations, such as non-real-time dynamic mapping and relatively coarse spatial resolution, they are the most accurate, validated, and widely used datasets in the region. That is why we used them for comparison and analysis with our near-real-time water surface dataset.

This work presents (i) the mapping of the water surface in the Rio Negro River basin at a 10 m spatial resolution and a 12-day temporal resolution using Sentinel-1 images and deep learning; (ii) a comparison of our water surface mask with the Joint Research Centre (JRC) global surface water (GSW) at a 30 m resolution, with the MapBiomas Water initiative water surface in 2022 at a 30 m resolution, with the LBA-ECO LC-07 dataset of wetland extent and vegetation types, and finally, with the water level at the Port of Manaus; and (iii) a description of the Rio Negro River contraction during the 2023 drought event.

Water surface occurrence and recurrence in the 2022–2023 period are available at https://doi.org/10.5281/zenodo.10552959 (accessed on 15 March 2024).

## 2. Materials and Methods

### 2.1. Study Site

The Negro River Basin is among the largest tributaries of the Amazon River, ranking as the fifth largest river globally in terms of mean annual discharge [37]. It covers 696,000 km$^2$ (approximately 12% of the Amazon basin) in the countries of Brazil, Venezuela, Guyana, and Colombia (Figure 1) [3,37,38]. Black river water, typical of the Negro River basin, is attributed to its high content of dissolved organic matter and low sediment load [37–40]. Mean annual precipitation rates range from less than 2000 mm·yr$^{-1}$ in the northern part of the Branco River basin to up to 3000 mm·yr$^{-1}$ in the northwest [41]. The timing of the rainy season differs widely along the south-to-north gradient: the beginning of the rainy season occurs in December in the south and in March or April in the north, whereas the rainy period ends from May to October [38,42]. High water levels mainly occur from May to July, while low water levels occur from October to December, as measured at the Rio Negro River gauge at the port of Manaus (Brazil), https://www.portodemanaus.com.br/?pagina=niveis-maximo-minimo-do-rio-negro (accessed on 5 January 2024).

### 2.2. Sentinel-1 Satellite Images of the Rio Negro Basin

We used Sentinel-1 images from the global Sentinel-1 Ground Range Detected (GRD) archive at a ∼10 m spatial resolution, covering the Rio Negro Basin in 2022 and 2023 (Figure 1). The processing level was Ground Range Detected with High Resolution (GRDH). These images were acquired in Interferometric Wide Swath (IW) mode with the VV and VH polarizations (dual-band cross-polarization, vertical transmit/vertical receive, and vertical transmit/horizontal receive). VV polarization is sensitive to the roughness of the surface, with smoother surfaces generally resulting in lower backscattering values. VH polarization, on the other hand, is particularly sensitive to volume scattering, which is often associated with vegetation and water. Water surfaces typically exhibit low backscattering values in both VV and VH polarizations due to their smoothness, which results in specular reflection [9,10]. This specular reflection causes most of the radar energy to be reflected away from the sensor, leading to low backscattering values in both polarizations. Since the dataset

was not contiguous over the basin, we retained only the largest contiguous part as our study area (∼695,912 km$^2$, representing >95% of the basin area) (Figure 1). Sentinel-1 was accessed on 24 December 2023 and 3 January 2024 from https://registry.opendata.aws/sentinel-1/, and all images from the dates of 2022 and 2023 over the study area were downloaded (1536 images).

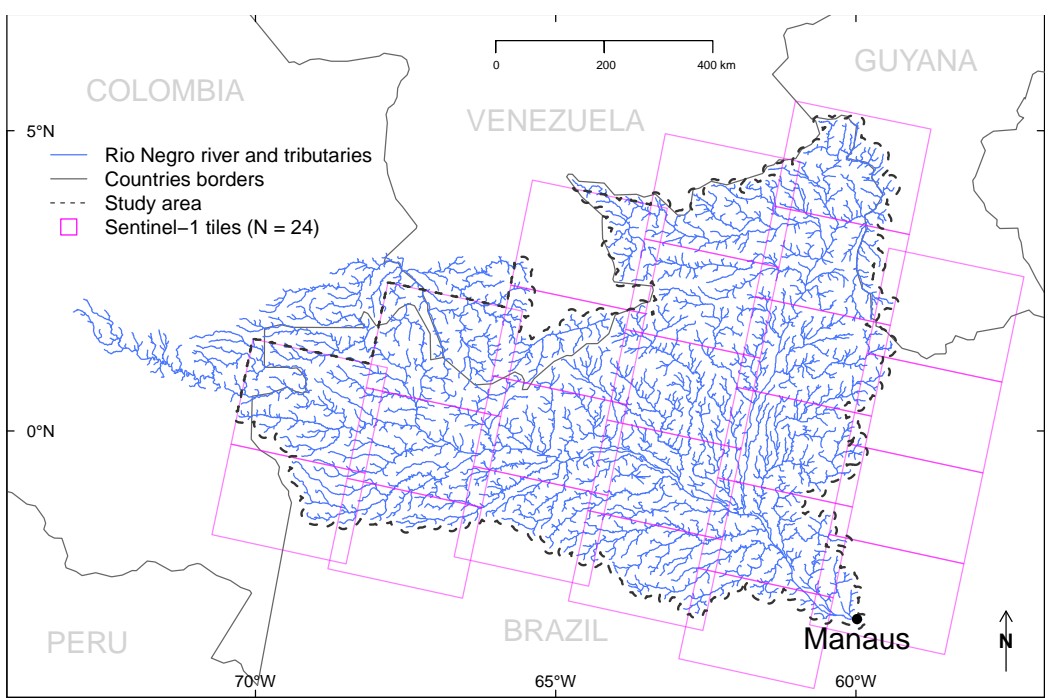

**Figure 1.** Geographical location in blue of the entire Rio Negro River and its tributaries before it forms the Amazon with the Solimões river at Manaus, adapted from [43]. The study area is represented by a dashed black line, and the approximate extents of the 24 Sentinel-1 images taken for each orbit are in magenta.

To obtain calibrated and georeferenced images suitable for our water segmentation model, the VV and VH images in Geotiff were pre-processed using the SNAP-Python (snappy) API (SNAP, ESA Sentinel Application Platform v9.0.0, http://step.esa.int (accessed on 15 November 2023)) of the Sentinel-1 Toolbox. The following steps were applied: (i) orbit correction to properly orthorectify the image using orbit metadata; (ii) GRD border noise removal to eliminate low-intensity noise and invalid data on scene borders, crucial for working with the entire scene; (iii) thermal noise removal to minimize discontinuities between sub-swaths in scenes acquired using multi-swath acquisition modes; (iv) calibration of the data to sigma-naught values, that is, computation of backscatter intensity with the sensor calibration parameters from the metadata; (v) terrain correction (orthorectification) using Global Earth Topography And Sea Surface Elevation at a 30 arc-second resolution (GETASSE30) as the digital elevation model. The terrain-corrected values were converted to decibels via log scaling (10×log10(x)), and the VV and VH image values ranging between −49 and 1 were scaled to 0–255 for storage as an 8-bit integer 2-band raster, following the formula (maximum (−49, minimum (1,x)) + 50) × 5.

## 2.3. Sentinel-1 Mountain Shade Mask from SRTM

Topographic features create shadows in the SAR imagery that exhibit similar backscatter values to the water surface and may be misclassified by the model. Consequently, to exclude shaded areas from our results, we generated a shade mask using the Shuttle Radar Topography Mission (SRTM) data at 30 m [44] to identify mountains and shaded slopes. The SRTM data for the study area were downloaded on 29 December 2023 from https://dwtkns.com/srtm30m/. The tiles were mosaicked into a single image, and aggre-

gated by a factor of 3 (90 m spatial resolution) using the minimum value using GDAL [45]. Then, slope computation was performed by considering the eight neighboring pixels with the R package `terra` [46]. A value of 1 was assigned to slopes over 20 degrees, and 0 otherwise. The mask was then converted to polygons [47,48]. Holes in the polygons were removed [49], and the convex hull envelope was computed to ensure coverage of all mountains. Minor polygon errors (shaded areas over rivers) were manually edited. Finally, these polygons were rasterized into images matching the size and resolution of the images receiving the results of water segmentation.

### 2.4. High-Resolution Water Masks and Water Levels

To compare and analyze our water segmentation results, we used four datasets independent from Sentinel-1 observations over the region (Table 1). The first product is the Joint Research Centre (JRC) global surface water (GSW) at a 30 m resolution based on Landsat data [4]. From the GSW product, we extracted data on water occurrence (percent of observation of the pixels representing water in the time series, where 100 indicates constant water) and recurrence (the frequency of water reappearing from year to year across the time series) over the Rio Negro basin. The datasets were accessed on 21 December 2023 from https://global-surface-water.appspot.com/download. The second product is the annual water surface of 2022 over our study area from the MapBiomas Water initiative at a 30 m resolution based on Landsat data [36] (MapBiomas Project-Collection 8 of the Annual Land Use Land Cover Maps of Brazil, accessed on 13 January 2024 through the link: https://brasil.mapbiomas.org/colecoes-mapbiomas/). The water surface for the year 2022 (annual_water_coverage_2022) was selected from the asset mapbiomas_water_collection2_annual_water_coverage_v1. The third product is the LBA-ECO LC-07 Wetland dataset [5] accessed from https://daac.ornl.gov/cgi-bin/dsviewer.pl?ds_id=1284 (accessed on 1 December 2023). This dataset consists of maps of wetland extent, vegetation type, and dual-season flooding state of the entire lowland Amazon basin at 3 arc-seconds of spatial resolution (~90 m). These maps were derived from mosaics of Japanese Earth Resources Satellite (JERS-1) synthetic aperture radar (SAR) imagery for the period October–November 1995 and May–July 1996. Full class and values descriptions are available at https://daac.ornl.gov/LBA/guides/LC07_Amazon_Wetlands.html (accessed on 1 December 2023). The fourth product is the daily Rio Negro water level measured at the port of Manaus (Brazil), accessed on 6 January 2024 from https://www.portodemanaus.com.br/?pagina=nivel-do-rio-negro-hoje.

**Table 1.** Satellite-based products and gauge measurements used in this study and their characteristics.

| Products | Satellite Data | Spatial Resolution | Temporal Resolution | Reference |
|---|---|---|---|---|
| Water masks for training | Planet NICFI derivative | 4.78 m | Monthly | This paper |
| Our water surface model | Sentinel-1 | 10 m | 12 days | This paper |
| Global Surface Water (GSW) | Landsat Data Archive | 30 m | Multiyear | [4] |
| MapBiomas Water Initiative | Landsat Data Archive | 30 m | Yearly | [6,7,36] |
| LBA-ECO LC-07 | JERS-1 SAR | 90m | November 1995/July 1996 | [5,35] |
| SRTM | C-band SAR (Space Shuttle) | 30 m | February 2000 | [44] |
| Rio Negro water level | — | Port of Manaus | Daily | https://www.portodemanaus.com.br/?pagina=nivel-do-rio-negro-hoje |

### 2.5. Model Architecture

The water surface segmentation was performed using a classical U-net model [29] (Figure 2). Specifically, the U-net model returns the probability of water surface presence in each pixel of a given input image. The model takes 2-band (VH and VV) Sentinel-1 images with a size of 256 × 256 pixels as inputs. The output is a one-band mask with dimensions of 256 × 256 pixels, containing 1 (indicating a water surface pixel with a probability >=0.5)

or 0 (representing a non-water pixel with a probability <0.5). The model was implemented in the R language [50] using the RStudio interface to Keras and TensorFlow 2.10 [51–54].

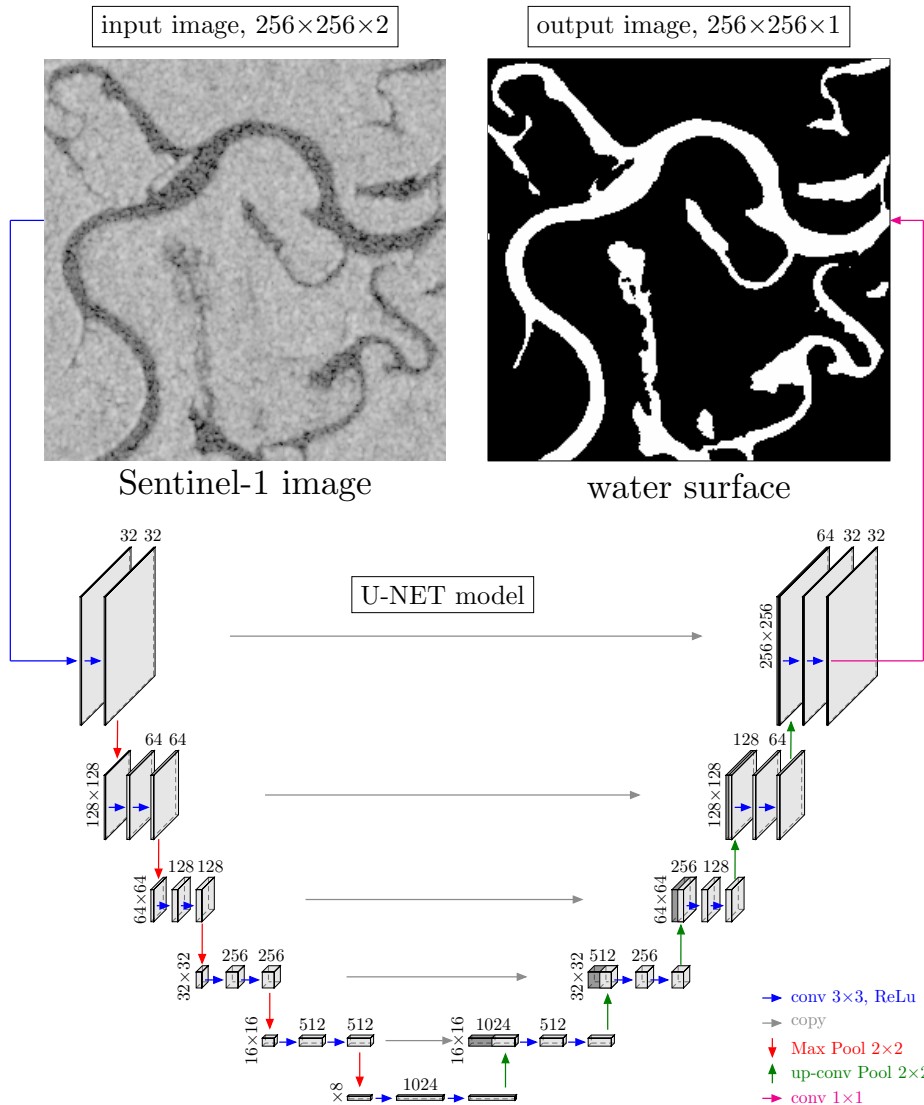

**Figure 2.** U-Net model architecture used for water surface estimation from Sentinel-1 images, adapted from Ronneberger et al. [29]. The number of channels is indicated above the cuboids and the vertical numbers indicate the row and column size in pixels. The operations (convolutions, skip connections, max pooling, and upsampling) performed in each layer and their sizes are indicated by the colored arrows.

## 2.6. Network Training

To generate training samples for the U-net model's water surface, we used a water mask at a 4.78 m resolution derived from a previous model using Planet NICFI [55,56]. While this dataset based on Planet NICFI has a higher spatial resolution than Sentinel-1, there are many clouds in the NICFI images over this region, making it unsuitable for near-real-time mapping. From this dataset covering Amazon water bodies from 2015 to 2023 and tiled with the Planet NICFI tiling system, we selected water masks from 2021 to 2022 that were fully covered by a Sentinel-1 image (acquired ≤25 days apart from the NICFI mosaic date) and had less than 25% cloud cover. Then, water masks were randomly selected for each monthly Planet mosaic date from 2021 and 2022, to constitute a data set of 550 non-overlapping images. Additionally, 29 Planet NICFI-derived water masks containing sandbanks in 2021 and 2022 were added to this dataset. This strategy was

implemented to have images for all seasons, use only non-overlapping scenes to prevent overfitting, and include sandbanks observed solely in the dry season (Figure 1).

The Sentinel-1 images corresponding to the water mask tiles were pre-processed as in Section 2.2 and clipped to the water mask extents. Subsequently, the water mask at a 4.78 m resolution was projected onto a raster with the same resolution and extent as the preprocessed and clipped Sentinel-1 image. Both the clipped Sentinel-1 image and the corresponding water mask, with the same size and resolution, were then cropped to a size of 1792 × 1792 pixels and retiled to the size of the 256 × 256 images for use in model training. The complete sample comprised 59,387 and 6057 Sentinel-1 image patches of 256 × 256 pixels and their associated water masks for training and validation, respectively.

Each image patch underwent a data augmentation process involving random vertical and horizontal flips. No additional data augmentation was necessary due to the natural data augmentation provided by the acquisition of sampling images on different dates [57]. Following data augmentation, the images were fed into the U-net model.

During network training, we used standard stochastic gradient descent optimization. The loss function was designed as the sum of two terms: binary cross-entropy and Dice coefficient-related loss of the predicted masks [51,52,58]. Finally, the optimizer Adam [59] with a learning rate of 0.0001 was used. We used the accuracy (i.e., the frequency with which the prediction matches the observed value) as the metric to assess the model performance.

The network was trained for 2000 epochs with a batch size of 32 images, and the model with the best accuracy was kept for prediction. The training of the models took approximately 4.5 h per 100 epochs using an NVIDIA A10G Tensor Core (GPU) with a 24 GB memory (NVIDIA Corporation, Santa Clara, CA, USA).

### 2.7. Prediction

For the prediction of the water mask, the 1536 Sentinel images (64 dates spanning from 1 December 2022 to 31 December 2023 for each of the 24 Sentinel-1 tiles (Figure 1)) were re-sized by adding columns and rows to have an aspect ratio of 4096 and a 128-pixel border. This adjustment was made to meet the input size requirements for prediction. The standardized-size Sentinel-1 tiles were then subdivided into sub-images of 4224 × 4224 pixels with a 128-pixel overlap between them. The total number of sub-images to predict was 69,577. Predictions were made on these 4224 × 4224 pixel images, and the 128-pixel border on each sub-image prediction was removed to mitigate border artifacts [29]. To create the 11-day water mask mosaics, the resulting 4096 × 4096 images were projected onto a regular grid with a tiling system of 4096 × 4096 pixel images, maintaining the same resolution as the central scene of Sentinel-1 over the study area during each 11-day period. In cases of overlap between predicted images, the maximum value was retained, and in the absence of data, the previous mosaic values were used to complete the time series. The final dataset, covering the period 2022–2023, consisted of 62 mosaics with a temporal resolution of 12 days. The computation time for predicting Rio Negro water masks using an RTX4090 GPU was 3.75 days.

### 2.8. Filtering for Artifacts in Sentinel-1 Images

Although the Sentinel-1 C-band SAR can theoretically penetrate clouds, some images may still exhibit significant artifacts, potentially from convective clouds, as previously observed in this region [60]. This leads to an increase in the values of the VV and VH bands, making it challenging for the model to detect water. To address this artifact in the prediction, we selected all tiles in our prediction grid with more than 500,000 pixels of river (68 tiles) based on a water occurrence mask computed for the entire time series (62 dates). For each of these 68 tiles, we identified the 5 images with the largest anomalies in pixels classified as water in the water occurrence mask and with fewer than two dates without water. Subsequently, all selected prediction tiles underwent visual inspection, and in cases where significant omissions of water pixels were clearly identified as errors, corrections

were made using the previous water mask mosaic date. A total of 111 tiles in our prediction grid were corrected for these omission errors, which represents less than 0.2% of the tiles.

*2.9. Segmentation Accuracy Assessment*

To evaluate the accuracy of segmentation on the training and validation samples, the F1-score was computed for the binary class $i$ = water surface as the harmonic average of the precision and recall (Equations (1)–(3)). The precision was the ratio of the number of segments classified correctly as $i$ and the number of all segments (true and false positives), and recall was the ratio of the number of segments classified correctly as $i$ and the total number of segments belonging to class $i$ (true positives and false negatives). This score varies between 0 (lowest value) and 1 (best value).

$$precision_i = \frac{true\ positive_i}{(true\ positive_i + false\ positive_i)} \tag{1}$$

$$recall_i = \frac{true\ positive_i}{(true\ positive_i + false\ negative_i)} \tag{2}$$

$$F1_i = 2 \times \frac{precision_i \times recall_i}{(precision_i + recall_i)} \tag{3}$$

## 3. Results

*3.1. Model Accuracy*

Overall, the F1-score, precision and recall values were similar for the training and validation samples (Table 2). The water surface model F1-scores of the predictions on the training and validation samples were both high, with a value of 0.930 (Table 2). The recall was lower than the precision, indicating a slightly higher rate of false negatives than of false positives.

**Table 2.** F1-scores of the water surface segmentation in the training and validation samples.

| Model | Sample | Images | Precision | Recall | F1-Score |
|---|---|---|---|---|---|
| Water mask | Training | 59,387 | 0.935 | 0.926 | 0.930 |
| | Validation | 6057 | 0.934 | 0.926 | 0.930 |

The model performs well on the validation dataset and can accurately capture most of the variations in the river border (Figure 3(a1–r2)). It can effectively segment very small islands (Figure 3(f2,g2)). However, the resolution of the mask may be slightly degraded in certain areas, as seen in Figure 3(o2,p2,q2). It is important to note that the water surface masks for training and validation come from a model at a 5 m spatial resolution (and were aggregated at 10 m), while the Sentinel-1 spatial resolution is 10 m. Overall, the model performs very well for non-water pixels and has very few commission errors. However, for very small rivers, there are some omissions, even in the validation images with the highest F1-score, where small rivers measuring one or two pixels in width (10–20 m) are missed (Figure 3(b2,h2,i2,j2)).

For the images in the validation with low F1-scores, errors can arise from the omission of small rivers with widths less than 10 to 20 m (Figure 4(a1–b2)). An inundation event in flooded shrubs was also missed by the model (Figure 4(c1,c2)). Errors from clouds are visible in Figure 4(d2,e2). These cloud errors are recognizable as their borders are uncorrelated with the river pattern and usually present in the entire Sentinel-1 image. Finally, some errors are due to clouds in the Planet NICFI-derived water surface mask (Figure 4(f2)), while water surface is correctly mapped from the Sentinel-1 image.

Concerning the errors observed in the prediction, common omission errors include the impact of clouds in the Sentinel image, resulting in an increase in VH and VV bands, leading to the omission of water. We filtered out most of these large errors in post-processing (see

Section 2.8). The band of water directly in front of the city of Manaus exhibits some omission errors, but it is challenging to determine if they come from the raw image, image correction, or proximity to very high VH and VV values in the city that could influence the model. This issue may be mitigated in the future near cities by incorporating training samples in those areas. In our study, this affects a negligible area in relation to the total water surface of the Rio Negro River. Regarding commission errors, all mountain-shaded areas are predicted as water, necessitating masking through alternative means, as they exhibit the exact same pattern as water (see Section 2.3). Additionally, a less significant commission error in terms of area over our study area, but consistently present, is airport landing runways.

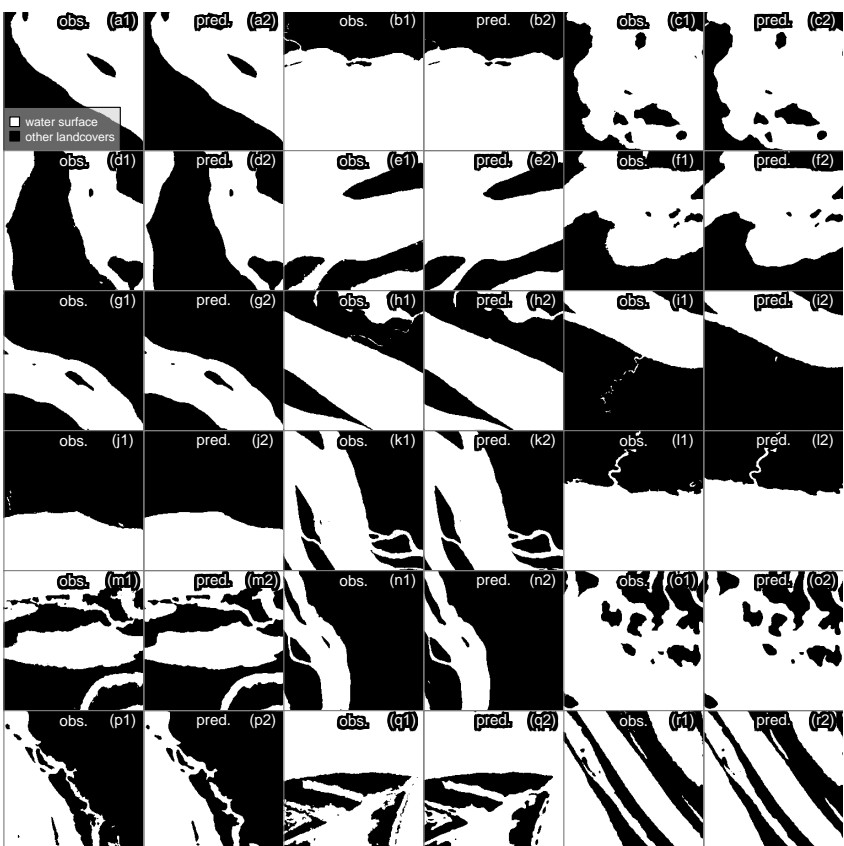

**Figure 3.** Observations and predictions for 18 images of the validation dataset with an F1-score above 0.97. For each location, the observed water mask is on the left and the predicted water mask is on the right.

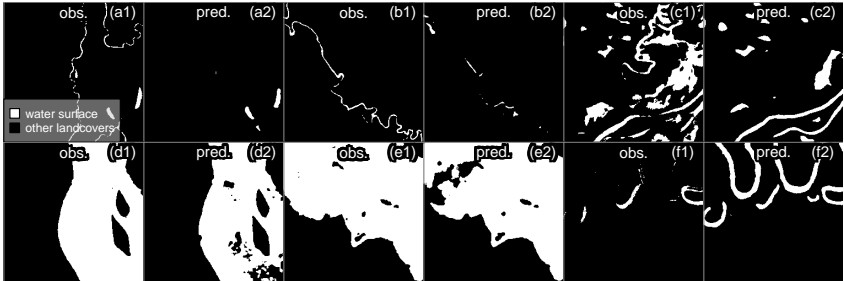

**Figure 4.** Observations and predictions for six images of the validation dataset with an F1-score below 0.5 or cloud artifacts. For each location, the observed water mask is on the left and the predicted water mask is on the right.

*3.2. LBA-ECO LC-07 Wetland Dataset*

Regarding the vegetation types and seasonal flooding states (wetland LC-07 data) of the pixels classified as water surface by our model (Table 3), 29.7% fall into the open water class in the wetland LC-07 data. The next most frequent classes were "Non-flooded shrub/Flooded shrub" (12.7%), Non-wetland within Amazon Basin (12.0%), "Non-flooded forest/Flooded forest" (11.3%), and "Flooded woodland/Flooded woodland" (8.1%). The high percentage of pixels classified as "Non-wetland within Amazon Basin" (12.0%) is likely due to the higher spatial resolution (10 m) of our dataset, which maps more smaller rivers than the LC-07 wetlands dataset (90 m spatial resolution).

**Table 3.** Frequency and percentage of Wetland classes for the pixels classified as water by our model and for the pixels classified as non-water by our model but classified as water surface in the GWS dataset (false negative).

| Vegetation Type and Dual-Season Flooding State from LBA-ECO LC-07 Data | | Our Model | False Negative GWS | False Negative MapBiomas |
|---|---|---|---|---|
| Cover at Low Water Stage | Cover at High Water Stage | Freq. (%) | Freq. (%) | Freq. (%) |
| Non-wetland within Amazon Basin | Non-wetland within Amazon Basin | 3,040,627 (12) | 310,982 (10.2) | 743,987 (9.6) |
| Open water | Open water | 7,564,727 (29.7) | 338,274 (11.1) | 256,448 (3.3) |
| Open water | Aquatic macrophyte (flooded herbaceous) | 214,580 (0.8) | 8275 (0.3) | 22,046 (0.3) |
| Non-flooded bare soil or herbaceous | Open water | 894,024 (3.5) | 317,612 (10.4) | 392,493 (5.0) |
| Non-flooded bare soil or herbaceous | Aquatic macrophyte (flooded herbaceous) | 590,643 (2.3) | 199,393 (6.5) | 343,246 (4.4) |
| Aquatic macrophyte (flooded herbaceous) | Aquatic macrophyte (flooded herbaceous) | 366,235 (1.4) | 121,902 (4) | 223,821 (2.9) |
| Non-flooded shrub | Open water | 48,268 (0.2) | 2645 (0.1) | 1290 (0.0) |
| Non-flooded shrub | Flooded shrub | 3,239,111 (12.7) | 499,891 (16.4) | 1,700,976 (21.9) |
| Flooded shrub | Open water | 496,599 (2) | 206,689 (6.8) | 318,729 (4.1) |
| Flooded shrub | Flooded shrub | 25,446 (0.1) | 3748 (0.1) | 4115 (0.1) |
| Non-flooded woodland | Flooded woodland | 501,838 (2) | 30,312 (1) | 17,367 (0.2) |
| Flooded woodland | Flooded woodland | 2,059,197 (8.1) | 619,445 (20.3) | 2,003,905 (25.7) |
| Non-flooded forest | Non-flooded forest | 1,692,959 (6.7) | 136,088 (4.5) | 697,255 (9.0) |
| Non-flooded forest | Flooded forest | 2,863,932 (11.3) | 94,924 (3.1) | 343,367 (4.4) |
| Flooded forest | Flooded forest | 1,723,735 (6.8) | 147,859 (4.9) | 632,938 (8.1) |
| Elevation >= 500 m, in Basin | Elevation >= 500 m, in Basin | 105,759 (0.4) | 10,394 (0.3) | 81,478 (1.0) |

### 3.3. Comparison with MapBiomas Water Initiative Data—Year 2022

We found 24,761,019 and 24,774,540 pixels, corresponding to ~22,285 and ~22,297 km$^2$ of water surface at a 30 m resolution from our model in 2022 and from the MapBiomass water surface in 2022, respectively. The F1-score between water presence/absence in the MapBiomas dataset and our water surface at the same spatial resolution (30 m) was 0.686. The precision was 0.686, indicating a high rate of false positives, which was expected, as our dataset was produced at a higher resolution (10 m), allowing us to capture more rivers. The recall was 0.686, similar to the precision value, indicating a significant rate of false negatives. The main differences can be found in the vegetation and seasonal states of the pixels classified as false negatives (Table 3). Almost half of the false negatives fell in the classes annually flooded woodland (25.7%) and seasonally flooded shrub (21.9%).

### 3.4. Comparison with Global Water Surface (JRC)—Period 1984–2015

We found 25,428,448 and 18,650,053 pixels (~22,886 and ~16,785 km$^2$) of water surface at a 30 m resolution from our model in the period 2022–2023 and the GWS dataset, respectively. The F1-score between water presence/absence in the GWS dataset and our water surface at the same spatial resolution (30 m) was 0.708. The precision was 0.614 in relation to the GWS dataset, indicating a high rate of false positives, as expected, since our dataset was produced at a higher resolution (10 m), allowing us to capture more rivers. The recall was 0.836, indicating a relatively low rate of false negatives. As for the comparison with the MapBiomas water surface, a product that is also based on Landsat data, the two prominent classes in the false negatives were annually flooded woodland (20.3%) and seasonally flooded shrub (16.4%).

### 3.5. Regional Results and 2023 Drought at 10 m Spatial Resolution

The correlation between the Rio Negro River water levels measured at the Port of Manaus and the water area for the basin, measured from our model and Sentinel-1 images, was high, with a value of 0.887, as shown in Figure 5. The minimum and maximum values of both curves always occurred with a maximum delay of ~ one month. For the highest water level around July, the water surface is more variable, likely due to large new water surfaces from inundation. For the lowest water levels, both curves exhibit a similar pattern of variation, suggesting that water surfaces are more associated with the river's water level during the dry period of the year.

The median water surface area for the 2022–2023 period was 11,795.2 km$^2$. The maximum water surface was observed on 9 July 2022, reaching an area of 14,036.3 km$^2$. The lowest water surface, 9559.9 km$^2$, was recorded on 25 November 2023, in the water surface mosaic composed of Sentinel-1 images taken from the 14th to the 25th. During the 2023 drought, the water surface decreased to 68.1% of the maximum observed and 81.0% of the median observed during the period 2022–2023.

Two of the major archipelagos in the Rio Negro River, near the junction with the Rio Branco river and the Rio Jufari river (Figure 6a–c), and near the Anavilhanas Archipelago National Park (Figure 6d–f), showed a remarkable pattern of changes between the highest and lowest water surfaces estimated by our model. During the highest measured water surface, some islands are still above water but the river dominates the landscape, with 1055.6 km$^2$ for the Rio Branco junction (Figure 6a) and 1711.6 km$^2$ for the Anavilhanas Archipelago (Figure 6d). During the lowest measured water surface on 25 November 2023, the water surfaces were reduced by half in both archipelagos, with 549.5 km$^2$ (52.1% of the highest level) for the Rio Branco junction and 841.2 km$^2$ (49.1% of the highest level). The part of the river that disappears in Figure 6b still has some water during the drought, as verified in the Planet NICFI images at a 5 m resolution for October and November. However, it remains undetected by the model for the date of the lowest point and the three previous dates, and the river becomes connected again on the date after the lowest point. Also, the VV and VH bands look more similar to sand banks than to water. This could be explained by thin water paths that are undetected, very low water levels affecting

radar measurements (shoaling), or artifacts of the GRD border noise removal to eliminate low-intensity noise and invalid data on scene borders in the image, as this river is on the overlap of two Sentinel-1 orbits.

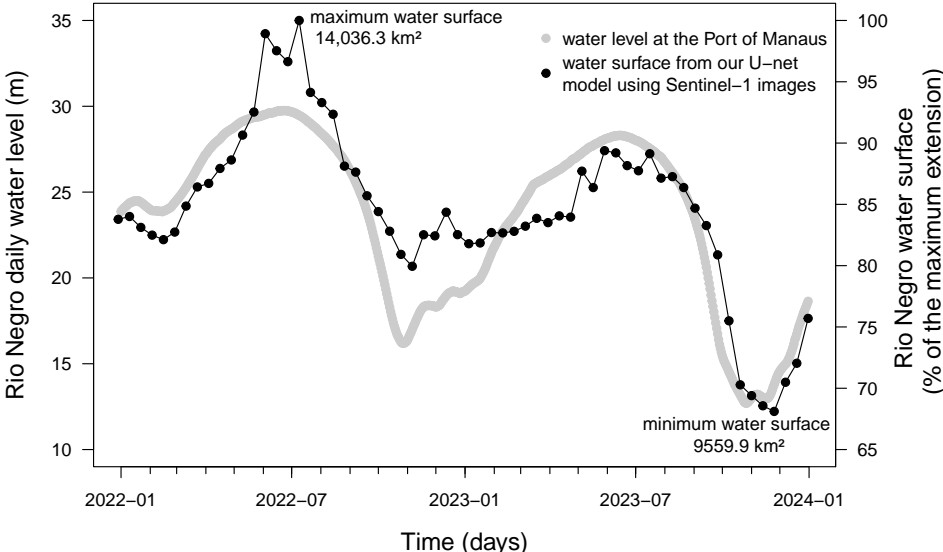

**Figure 5.** Daily water level of the Rio Negro measured at the Port of Manaus and water surface estimated every 12 days with our Sentinel-1-based model for the Rio Negro study area, for the period 2022–2023.

For both the junction with the Rio Branco river and the Anavilhanas Archipelago, 40% of the water surface is permanent (occurrence of 100%) (Figure 6c,f). However, the seasonality differs, as the value of the 20th percentile for water occurrence at the river junction with the Rio Branco river is 54%, while for Anavilhanas Archipelago, it is 85%. This indicates less seasonality in water surface for the Anavilhanas Archipelago, and there is a greater possibility of inundation at the junction with the Rio Branco river, as seen with reddish colors in Figure 6c. An artifact indicating omission errors at the border between two images in the same orbit is highlighted by a white arrow in Figure 6f. There is no overlap between images captured in the same orbit, and this artifact is likely attributed to a border effect inherent in the raw Sentinel image.

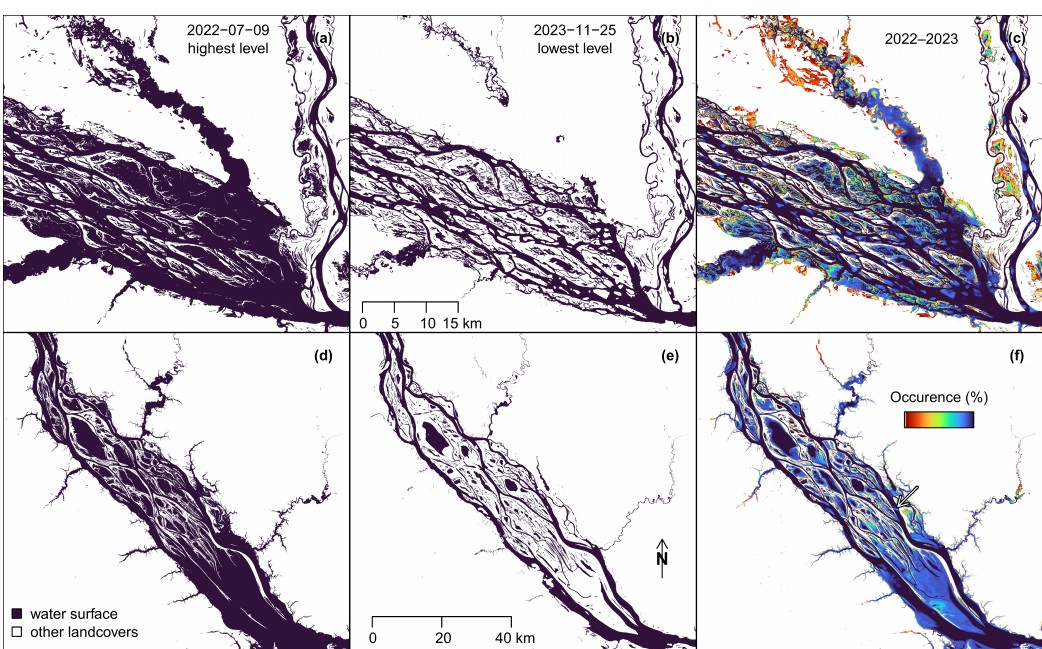

**Figure 6.** Examples of our deep learning water segmentation results based on Sentinel-1 images for the Rio Negro water surface near the junction with the Branco River (**a**–**c**) and near the Anavilhanas Archipel National Park (**d**–**f**), for the maximum observed (9 July 2022, first column), the minimum observed (2023 drought, 25 November 2023, second column), and water occurence per pixel in % computed in the 2022–2023 period (third column). A white arrow in (**f**) indicates omission errors at the border between two images in the same orbit.

## 4. Discussion

### 4.1. Mapping Water Surface of the Rio Negro River and Perspectives

Here, for the first time, we demonstrate that medium-spatial-resolution Sentinel-1 C-band SAR allows the accurate estimation of water surface extent and changes in the Rio Negro River in near-real time, with a 12-day temporal resolution and 10 m spatial resolution. The water surface segmentation model achieved an F1-score above 93% on the validation dataset, highlighting once again the high capacity of deep learning to support water mapping in tropical environments [28,30–32]. The good performance of the segmentation could be explained by the unique water surface low backscattering value in comparison to other land cover types [9,10]. Our model enables the accurate near-real-time monitoring of water surfaces and performed particularly well during drought. Furthermore, the model appears to be robust to temporal variations in Sentinel-1 images, as the validation images are distributed relatively evenly across all months. Apart from SAR's ability to penetrate through clouds, which allows for mapping even under the persistent cloud cover common in this region, Sentinel-1 has several advantages for water level monitoring. Its 12-day acquisition frequency enables the rapid monitoring of water levels, as they can change quickly, as shown in Figure 5. Additionally, its high resolution enables the mapping of smaller rivers and river borders with more detail than existing products. For the period 2022–2023, we estimated a median water surface area of 11,795.2 km², a maximum of 14,036.3 km², and a minimum of 9559.9 km², observed during the 2023 drought (Figure 5). In this major drought, with water levels reaching the lowest observed since 1903, the water surface decreased to 68.1% of the maximum observed and 81.0% of the median observed during the period 2022–2023. When water levels are at their lowest during the year, the water level and water surface exhibit a similar pattern of variation, suggesting that water surfaces are more associated with the river's water level during the dry period of the year. This drought has a negative effect on human communities that rely on the river for basic resources such as water, food, and transportation, including medical care and education [61]. It also affects natural ecosystems, which face unusual conditions, such as

an increase in water temperature that has likely caused the death of several endangered Amazon river dolphins [62]. Extreme climate events, such as major floods and droughts, have become more frequent in the Amazon region in the last few decades [63–65], and our model could be used to monitor near-real-time changes in water extents in this region. Furthermore, in combination with recent data from the SWOT mission, which observes surface water storage change and fluxes on a global scale [66], it could be used to model the real water balance of the Amazon River, in addition to its water surface extent. Finally, in a broader context, as flooding affects large regions in the world each year [24,67], our model could be tested for the near-real-time mapping of flood events.

### 4.2. Limitations of the Water Surface Model

The model made very few commission errors after masking for mountain shades. Although they do not represent a significant area, airport runways and large roads (∼four lanes) were predicted as water surfaces, likely due to low backscattering values in polarization, resulting from specular reflection over smooth surfaces, as observed in the SAR images [68]. Most mountain shades were predicted as water surfaces due to similar backscattering values. We could have used SRTM at a 30 m spatial resolution for terrain correction, which is the conventional way. However, it was faster to process the terrain correction of the 1536 images using the Global Earth Topography And Sea Surface Elevation at a 30 arc-second resolution (GETASSE30) as the digital elevation model. Furthermore, most rivers were located at low altitudes and far from mountain relief that could have caused shadows. Opting for an independently created and manually corrected shadow mask was faster, taking under half a day, than performing SRTM terrain correction at 30 m in preprocessing and addressing shadows within the model. Finally, although there are still a few shadows present on the map, they are not supposed to change in area during the year and impact the seasonality of the water surface, as acquisitions are the same throughout the year for an orbit.

Omission errors were more common than commission errors, and we identified two main sources: clouds and seasonally flooded shrubs and trees. Sentinel-1 C-band SAR can penetrate most clouds, but some images showed significant artifacts (less than 0.2% of the tiles of our prediction grid, Section 2.8), potentially from convective clouds, as already observed in the Amazon region [60]. This leads to an increase in the values of the VV and VH bands, making it challenging for the model to detect water. Unlike multi-spectral satellite images, clouds are not easily detected in the VH and VV bands, and only large errors could be detected and corrected when they were visible in the prediction mask (Section 2.8). This is likely the major problem of detecting water surfaces in near-real time. The second main source of omission errors was flooded vegetation, with ≥35% of the omission errors located in the classes 'annually flooded woodland' and 'seasonally flooded' of the LBA-ECO LC-07 vegetation type and dual-season flooding state dataset (Table 3). This might indicate that Sentinel-1 C-Band is sensitive to the remaining vegetation structure above water during the flood period. These omission errors are mostly located in the Rio Negro interfluvial wetlands, a region found 200 km north/north-west of the junction of the Rio Negro and Rio Branco rivers, only flooded during the wet season [3,5]. While this could be a problem for inundation mapping, it has no effect on estimating the river contraction during a drought. Studies that classify flooded forests and floodplain lakes with emergent shrubs with radar usually use more SAR bands and polarizations, not only C-band and VV or VH polarization, to benefit from the radar double-bounce returns from water and vegetation surfaces [9,10,69]. In 2024, the NISAR mission will be launched, providing SAR data (L-band, 24 cm wavelength, and S-band, 10 cm wavelength) distributed at a 3 m spatial resolution and a 12-day temporal resolution over land [70]. This new SAR data will likely complement the Sentinel-1 data to improve the mapping of flooded forests and floodplain lakes with emergent shrubs. There were a few omissions in the water front of Manaus, which could have originated from the raw image, insufficient training data near the city, image correction, or proximity to very high VH or VV values in the city that could impact

the model prediction. While this was not significant in our study, it could have an impact in regions with heavily urbanized river borders. Our approach, while mapping more rivers than GWS and MapBiomas, is limited to rivers with a significant width, typically 50 m or more, and smaller rivers detected inside the Amazon forest may take time to appear in the dataset due to the Sentinel-1 incidence angle. Finally, a few omissions were found in a band of approximately 200 m on the border between images of the same orbit (10 pixels of 10 m on each side). Contrary to neighboring orbits, these images had no overlap between them. This might be a border effect from the model, the raw data, or GRD correction. If it comes from the model, it might be resolved by filling the missing part of the image with the value in the neighboring image of the same orbit. Finally, the backscatter coefficients, especially the VV polarization, can also be influenced by wind-induced surface roughness over open water [27] and/or low water levels causing shoaling [71], and this might explain why water can disappear from the prediction (Figure 6b). However, it could also be due to a river width being too small to be detected or artifacts from GRD border noise removal, as this river is on the overlap of two Sentinel-1 orbits, and more investigation is needed.

*4.3. Water Surface Segmentation Model Application on Larger Scale*

While this study maps the water extent of one of the major Amazon River tributaries, it still represents only around 20% of the water extent of the Amazon basin [7], and only during 2 years. For the application of this method on a larger scale, temporally or spatially, there are three main challenges. The first is the dataset; Sentinel-1 data does not provide a ready-to-use product for the user, and each image has to be preprocessed by SNAP, which takes a significant amount of time (>30 s per image) and limits the application on a larger scale to a single user on a single machine. The second limitation is the absence of a standard Sentinel-1 tiling scheme, making it challenging for the user to easily find all images corresponding to their region and date of interest without having to search on external databases, which can be problematic for large datasets. The third limitation is that in Google Earth Engine [34], the only place where the Sentinel-1 images are preprocessed and ready to use, the use of deep learning, while already tested and working [28], would likely have a prohibitive cost for large regional/country scales. To conclude, users would benefit from an archive of ready-to-use preprocessed 12-day Sentinel-1 mosaics with a standard tiling scheme.

## 5. Conclusions

In this study, we developed a deep learning-based method to automatically map the water surface of the Rio Negro River and its changes using Sentinel-1 SAR images. We found that Rio Negro water surfaces during the October–November 2023 drought were reduced to 68.1% (9559.9 km$^2$) of the maximum water surfaces observed in the period 2022–2023 (14,036.3 km$^2$). The accuracy of our water surface model, demonstrated by an F1-score of 0.93 and comparison with reference products such as GSW and the MapBiomas Water initiative, as well as the Rio Negro River gauge at the Port of Manaus, allowed us to produce accurate 12-day maps of water surface at a spatial resolution of 10 m. Speckle noise was not removed from the raw images and does not appear to significantly impact the results of the U-net. However, rare clouds, likely convective clouds, can still produce omission artifacts in the water surface masks. These artifacts can be mostly corrected by temporal filtering. Our method required only minimal supervision and is relatively fast, consistent, and scalable. Given the expected increase in climate extremes in the Amazon region, our model could be used to monitor Amazon water surfaces in near-real time and, more broadly, tropical water surfaces.

**Author Contributions:** Conceptualization, F.H.W., S.F., R.D., M.C.M.H., A.M. and S.S.; methodology, F.H.W., S.F., R.D., M.C.M.H., A.M. and S.S.; software, F.H.W., S.F. and A.M.; validation, F.H.W. and S.F.; formal analysis, F.H.W. and S.F.; investigation, F.H.W., S.F., R.D., M.C.M.H., A.M. and S.S.; resources, S.S.; data curation, F.H.W., S.F. and M.C.M.H.; writing—original draft preparation, F.H.W., S.F., R.D., M.C.M.H., A.M. and S.S.; writing—review and editing, F.H.W., S.F., R.D., M.C.M.H., A.M. and S.S.; visualization, F.H.W.; supervision, S.S.; project administration, S.S.; funding acquisition, S.S. All authors have read and agreed to the published version of the manuscript.

**Funding:** This research received no external funding.

**Data Availability Statement:** Data on water surface occurrence and recurrence in the 2022–2023 period are available at https://doi.org/10.5281/zenodo.10552959 (accessed on 14 March 2024). Sentinel-1 data are available at https://registry.opendata.aws/sentinel-1/ (accessed on 14 March 2024).

**Acknowledgments:** The authors are grateful to the Grantham and High Tide Foundations for their generous gift to UCLA and grants to CTrees for bringing new science and technology to solve environmental problems. This work was partially conducted at the Jet Propulsion Laboratory, California Institute of Technology (under contract 80NM0018F0590), the National Aeronautics and Space Administration (NASA).

**Conflicts of Interest:** The authors declare no conflicts of interest. The funders had no role in the design of the study; in the collection, analyses, or interpretation of the data; in the writing of the manuscript; or in the decision to publish the results.

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
