# Peer review of "The Amazon’s 2023 Drought: Sentinel-1 Reveals Extreme Rio Negro River Contraction"

_remotesensing, doi:10.3390/rs16061056_

Round 1

Reviewer 1 Report

Comments and Suggestions for Authors

In this work, the authors seek to map water surfaces with a high spatiotemporal resolution (12 days, 10 m) derived from Sentinel-1 images using U-net deep learning model to demonstrate the effects of drought. A good performance of the segmentations and predictions is found with high F1-scores and the commission and omission errors are also justified by other reliable datasets. The idea sounds practical and interesting but the manuscript needs further refinement, especially in the Introduction and Discussion sections. My comments are focused on improving logical expression and standardability:

1.     Lines 74-98: The three datasets for comparison in this work have certain limitations, such as non-real-time dynamic mapping and relatively coarse spatial resolution. Whether the reliability of them in the relevant field has been proven? How they accurately represent the benefits of the outcome dataset based on Sentinel-1 and deep learning?

2.     Lines 101-102: The full name and abbreviation of the noun have been indicated in the preceding paragraph, and there is no need to repeat its full name in the following paragraph. Please check carefully and modify it.

3.     Line 109: It would be desirable that the authors add a textual description of the study site in this section.

4.     Line 111: The full name of a noun used as an abbreviation should be indicated when it first appears. Please check carefully and modify it.

5.     Lines 151-173: It is recommended that the authors add a summary table to clearly represent all the data used in this work and their attributes (e.g., acquisition, resolution and parameters).

6.     Line 247: Is the citation format to this equation here "eq. 1". Please check and modify it.

7.     Lines 362-386: I think what the authors need to focus on in this section are: what are the advantages of Sentinel-1 C-band SAR in monitoring the water surface changes and drought, and what is the necessity and practical significance of high spatiotemporal resolution images in this issue?

8.     Line 406: What images do the "111" here refer to? Please check and modify.

9.     Line 458: I think the authors should add the Conclusion section to make the manuscript complete and standardized.

Author Response

reply in the attached pdf

Reviewer 2 Report

Comments and Suggestions for Authors

The paper "Amazon's 2023 Drought: Sentinel-1 Reveals Extreme Rio Negro River Contraction" presents a study on mapping the water surfaces of the Rio Negro River basin in the Amazon using Sentinel-1 satellite radar images and deep learning techniques. The authors trained a U-net model to accurately segment water surfaces from the Sentinel-1 images and compared their results with existing water surface datasets. They also analyzed the contraction of the Rio Negro River during the 2023 drought event. T The paper presents an innovative methodology for mapping water surfaces using Sentinel-1 SAR data and deep learning techniques. The results are insightful and contribute to the field of remote sensing. However, there are some weaknesses in the implementation details, evaluation of the proposed method, and organizational structure. With improvements in these areas, the paper has the potential to be highly valuable to readers.

(1) Please provide the specific meaning of F1-score in the abstract.

(2) The paper lacks details on the implementation of the U-net model, such as the architecture, training parameters, and data augmentation techniques used. More information on these aspects would be helpful for reproducibility.

(3) The evaluation of the proposed method is limited to the F1-score, and there is no comparison with other state-of-the-art methods for water surface mapping. Including such comparisons would strengthen the paper's claims.

(4)The paper does not thoroughly discuss the limitations and potential sources of error in the U-net model's predictions. How robust is the model to variations in the Sentinel-1 images? A thorough analysis of the model's strengths and weaknesses would enhance the understanding of its applicability.

(5) The paper lacks a discussion of the potential applications and implications of the model's findings, such as the impact of the 2023 drought on the Rio Negro River.

(6) The paper does not explicitly present research conclusions.

Author Response

reply in the attached pdf

Reviewer 3 Report

Comments and Suggestions for Authors

Your paper is very interesting and worth to be published. I have some minor suggestions/questions.

1/ Sentinel-1 product you use is probably GRDH. Please make sure it has a spatial resolution of 10 m. I am not sure.

2/ It would be useful to explain the difference between VV and VH polarisations in the context of water detection which occurs as a mirror reflection with low reflectance. Have you been calculating some remote sensing indexes using VV and VH bands?

Author Response

reply in the attached pdf

Reviewer 4 Report

Comments and Suggestions for Authors

Dear authors and editors. 

I am grateful for the opportunity to review the scientific article Amazon's 2023 Draft: Sentinel-1 Reveals Extreme Rio Negro River Contract.

1. In the legend to the geographical map in Figure 1, specify the borders of the states. 

2. Why did you use SRTM and not Copernicus DEM (30 m/pixel)?

Others:

1. The article is dedicated to an important and interesting topic. The research discusses how the use of Sentinel-1 in conjunction with deep learning methods can improve the mapping of water surfaces in tropical regions almost in real-time, and the water surface mask demonstrated consistent correspondence to Global Surface Water and Mapbiomas products.

2. I believe the article presents an original research methodology that can be easily applicable to other regions worldwide. The authors thoroughly outline the research methodology, making the study particularly valuable in terms of reproducibility.

3. The article significantly expands the understanding of the chosen research area, as it explores a region that is difficult to access. The manuscript describes and interprets data from various sources, providing more reliable and up-to-date conclusions. The authors identify trends and patterns that were not previously observed or adequately studied, thereby enhancing knowledge about the selected area and its characteristics. Additionally, they develop methodological approaches and analytical tools that can be used for research in other regions or similar areas, making the article significant for a broader audience.

4. I find the research methodology to be well-written. Perhaps the authors should more thoroughly specify the limitations of the study and justify the choice of databases for analysis compared to those with higher spatial resolution. I believe the authors should add a "Conclusion" section to make the manuscript more complete.

5. I believe the results obtained by the authors align with the stated objectives and the presented research methodology. Geoinformation modeling and the use of training models have allowed for reliable and up-to-date results.

6. I find the references appropriate.

7. In the legend to the geographical map in Figure 1, please specify the borders of the states. No other comments regarding the other figures.

--

Author Response

reply in the attached pdf

Round 2

Reviewer 1 Report

Comments and Suggestions for Authors

At present the revised manuscript can be accepted for publication.

Author Response

Dear Reviewer,

Thank you very much for your work on the paper,

Best regards,

Fabien Wagner

Reviewer 2 Report

Comments and Suggestions for Authors

The revised version contains the FW character in many places. I don't understand what it means. Please check it.

Author Response

Dear Reviewer,

The FW character (my initials) appears in the track changes version of paper where the corrections have been made by me. It will not be present in the final version of the manuscript.

Thank you again for you work on the paper,

Best regards,

Fabien Wagner